# Do Perceptions about Palliative Care Affect Emergency Decisions of Health Personnel for Patients with Advanced Dementia?

**DOI:** 10.3390/ijerph191610236

**Published:** 2022-08-17

**Authors:** Meira Erel, Esther-Lee Marcus, Samuel N. Heyman, Freda DeKeyser Ganz

**Affiliations:** 1Henrietta Szold School of Nursing, Hadassah Hebrew University, Jerusalem 9112102, Israel; 2Herzog-Medical Center, Department of Geriatrics, Faculty of Medicine, Hebrew University of Jerusalem, Jerusalem 9103702, Israel; 3Department of Medicine, Hadassah Hebrew University Hospital, Mt. Scopus, Jerusalem 9765422, Israel; 4Faculty of Health and Life Sciences, Jerusalem College of Technology, Jerusalem 9372115, Israel

**Keywords:** advanced dementia, clinical decision-making, emergency treatment, health personnel, palliative care, terminal care

## Abstract

Decision analysis regarding emergency medical treatment in patients with advanced dementia has seldom been investigated. We aimed to examine the preferred medical treatment in emergency situations for patients with advanced dementia and its association with perceptions of palliative care. We conducted a survey of 159 physicians and 156 nurses from medical and surgical wards in two tertiary hospitals. The questionnaire included two case scenarios of patients with advanced dementia presenting gastrointestinal bleeding (scenario I) or pneumonia (scenario II) with a list of possible interventions and 11 items probing perceptions towards palliative care. Low burden interventions such as laboratory tests and intravenous administration of antibiotics/blood were preferred. Palliative measures such as analgesia/sedation were chosen by about half of the participants and invasive intervention by 41.6% (gastroscopy in scenario I) and 37.1% (intubation/mechanical ventilation in scenario II). Medical ward staff had a more palliative approach than surgical ward staff in scenario I, and senior staff had a more palliative approach than junior staff in scenario II. Most participants (90.4%) agreed that palliative care was appropriate for patients with advanced dementia. Stress in caring for patients with advanced dementia was reported by 24.5% of participants; 33.1% admitted fear of lawsuit, 33.8% were concerned about senior-level responses, and 69.7% were apprehensive of family members’ reaction to palliative care. Perceptions of health care workers towards palliative care were associated with preferred treatment choice for patients with advanced dementia, mainly in scenario II. Attitudes and apprehensions regarding palliative care in these situations may explain the gap between positive attitudes towards palliative care and the chosen treatment approach. Acquainting emergency care practitioners with the benefits of palliative care may impact their decisions when treating this population.

## 1. Introduction

Patients with advanced dementia (AD) are often referred from the community or long-term care facilities to general hospitals for the management of urgent medical conditions. The decisions of health personnel in these situations are often made under constraints of time and are often subject to uncertainty when there is no information regarding patients’ prior wishes or advance directives. When making treatment decisions for patients with AD, medical personnel are often required to consider the value of life versus the quality of life of their patients and adjust the care approach accordingly [1,2,3]. The terminal nature and trajectory of dementia, general perceptions, and attitudes towards palliative care (PC), the suitability of PC for this population [4], as well as the environment, often influence such decisions [5]. 

PC is both patient- and family-centered, and optimizes quality of life for people living with serious illness by anticipating, preventing, and treating suffering through the provision of relief from symptoms and the stress of the illness. PC throughout the continuum of illness involves addressing physical, intellectual, emotional, social, and spiritual needs and facilitates patient autonomy, access to information and choice [6]. 

Attitudes towards PC and its application to patients with AD have been extensively discussed, especially in the context of barriers to its use in this population, mostly in long-term-care facilities or community settings [7]. However, limited research has been conducted in acute care settings, specifically regarding emergency medical situations. 

Studies in acute settings show that ignoring PC for patients with AD is due to the perception that PC is not appropriate for this population [8]. There is a tendency to focus on the acute and critical state of the patient [9], and there are misconceptions related to PC [10]. Treatment approach is often not directed toward comfort until death is perceived as imminent [11]. 

In response to questions about the provision of end-of-life care for patients with AD, health personnel often report feelings of helplessness or low levels of self-confidence in the provision of PC [12,13]. Manu et al. [14] found that medical residents reported having lower levels of competence in the skills needed to care for patients with AD than for patients with metastatic cancer. Perceptions such as feelings of powerlessness or fear of becoming ill oneself may contribute to the perception of low levels of self-efficacy of health personnel providing end-of-life care for patients with dementia [12,13]. Some health personnel reported feeling a sense of failure and frustration when a patient dies, a desire to avoid taking care of patients with AD, or even distress and poor judgment in caring for patients with AD [15].

Under acute settings, a life-saving approach is usually considered the default treatment. Health personnel who consider care options that do not follow this approach may receive negative responses from their peers, which can lead to suppressing their initial intentions [16,17]. Factors associated with this approach are related to legal concerns, hierarchical and organizational authority, and responses of patient’s relatives [12,13,18].

The association between the perceptions of health personnel and the choice of appropriate treatment in patients with AD is especially relevant in the acute care environment. Hospital staff members have easy access to the most invasive and intensive interventions. Decisions related to which treatment option is chosen are often made under life-threatening, stressful, and emergency conditions. In such situations, healthcare providers are often unacquainted with the patient and the family, and do not know their treatment preferences [11,19]. These perceptions could impact the delivery of desired care for patients with AD under acute settings.

Our study was designed to examine the perceptions of acute care providers towards PC for patients with AD and to determine whether these perceptions are associated with their preferred decisions in acute immediate medical emergency situations. The study was based on questionnaires regarding decision-making in two common scenarios of life-threatening conditions in patients with AD. It included the use of scenarios that enable assessment of PC perceptions and treatment choices, while neutralizing the effects of the human environment in which medical care decisions are made. The selection of acute, urgent medical intervention is, therefore, an indicator of medical treatment preferences.

## 2. Materials and Methods

### 2.1. Design

This is a cross sectional survey, part of a mixed-methods (qualitative/quantitative) research study that evaluates preferences of physicians and nurses regarding treatment choices, including PC, for patients with AD under acute care settings. The qualitative part of the study included an interview assessing the thought processes involved in decision-making regarding a hypothetical case scenario of a patient with AD who presents with bowel obstruction related to a space-occupying lesion [18].

### 2.2. Participants

The study used a convenience sample of physicians and nurses who work in medical and surgical departments at two tertiary university-affiliated hospitals in Israel. These hospitals, although in different geographical regions, are comparable in size, and both serve patients from diverse ethnic and cultural backgrounds. Criteria for inclusion were medical personnel (physicians and nurses) without formal postgraduate training in geriatric medicine and/or PC. Participants were recruited during departmental staff meetings. We excluded health personnel with formal postgraduate training in geriatric medicine and/or PC since most of the personnel did not have such training, and this training may bias the findings of the study. Data collection took place from February 2018–February 2019 in Sheba Medical Center, and from July 2019–February 2020 in Hadassah Medical Center. The first author (M.E) participated in ward staff meetings, presented the study aims, and then administered the questionnaires.

### 2.3. Questionnaire

The questionnaire was comprised of two parts. The first part presented two hypothetical case scenarios describing patients with AD presenting with acute, urgent, potentially life-threatening medical situations, along with a selection of immediate medical treatment options for each scenario ranging from palliative to aggressive care. The scenarios presented were two common emergency health situations that included relevant medical intervention in older adults with AD. These options were chosen with the consent of all of the authors, and confirmed by six experts, who were board certified physicians in internal medicine and geriatric medicine working in acute care hospitals.

The first case scenario was a patient with gastrointestinal bleeding, and the second was an individual with aspiration pneumonia and acute respiratory failure (see Appendix A). The researchers asked participants to indicate interventions they would recommend for the patient from six to eight relevant choices. The interventions for scenario I were: urgent gastroscopy, insertion of a nasogastric tube, insertion of a central vein line, blood transfusion, intravenous infusion of fluids, laboratory tests, subcutaneous infusion of fluids, and analgesia. Interventions for scenario II included: intubation and mechanical ventilation, intravenous infusion of fluids, antimicrobial therapy, laboratory tests, analgesia, and sedation.

Each intervention was given a score within a range of −1 to +3, where more aggressive treatments received higher scores. For example, urgent gastroscopy received a score of +3, while PC treatments (such as analgesia or sedation) were awarded a score of −1 and a score of +1 if not chosen. An option other than PC intervention that was not chosen was given a score of 0. The sum of the scores of the chosen interventions, termed the “Palliative Score”, ranged from −1 to +17 and −2 to +11 for the first and second scenarios, respectively. Lower values reflected a more PC-oriented approach. Therefore, palliative interventions were given a negative value score of −1. The scores for the interventions were determined by consensus among the researchers and approved by the geriatric expert physicians. A pilot test on a subset of 19 participants was conducted and there was no need for further modifications in the case scenarios presented.

The second part of the questionnaire included 11 items addressing the perceptions of the participants related to PC for patients with AD. No valid and reliable questionnaire was found that met the study objectives when we conducted the study. Therefore, the questionnaire was based on a literature review that described the existing factors involved in providing PC for this population. It specifically addressed the assessment and thought processes involved in PC decision-making of healthcare staff in acute care settings regarding patients with AD and the potential barriers to providing PC [7,12,16,20].

The validation of the questionnaire items consisted of several steps. We first established the content validity, asking six physicians with expertise in this field of study to evaluate the questionnaire. The physicians were board certified in internal medicine and geriatric medicine; all worked in acute care hospitals. They evaluated whether the items effectively captured the topic under investigation. Following the pilot test mentioned above, no further modifications were made in this part of the questionnaire.

The 11 items included were: appropriateness of PC for patients with AD, colleagues’ perceptions of the appropriateness of PC, perceived ease of having an end-of-life conversation, death of a patient with AD perceived as failure and accompanied by guilt feelings, the ability to make care decisions for patients with AD, perceived stress and the desire to avoid decision-making for a patient with AD, four items assessing perceived legal and organizational concerns, and perceived concerns of responses of family members regarding PC approach. We also performed a principal component analysis that identified four underlying components representing the following themes: (1) apprehension, (2) ability to make decisions and being comfortable with end-of-life care, (3) appropriateness, and (4) fear of family member’s reactions. Using reliability statistics, a Cronbach Alpha on the standardized items reached 0.70.

Each item was graded using a five-point Likert scale. A lower score reflected a more positive perception of PC. All of the respondents completed the questionnaires during regular nursing or medical staff meetings. 

### 2.4. Ethical Considerations

The study was approved by the Institutional Review Board of each hospital (Sheba Medical Center, 4839-18-SMC; Hadassah Medical Center, 0027-19-HMO). Participants received oral and written information about the study, and they provided their written consent.

### 2.5. Data Analysis

SPSS version 25 (IBM SPSS Statistics for Windows, Armonk, NY, USA: IBM Corp.) was used for the data analysis. Descriptive statistics were applied to analyze study variables (participant characteristics, PC perceptions, and preferred medical treatments). Percentages were calculated for dichotomous and categorical characteristics. Mean, standard deviation, median, and range were presented for continuous variables. The first two perception items (PC appropriate for patients with AD; Perceived PC appropriate for patients with AD by colleagues) were scored as dichotomous perceptions (positive/negative), and the mean scores of the medical treatment choices (the Palliative Score) were compared using *t*-test and Wilcoxon non-parametric test for median comparison. The remaining nine perception items were scored as agree/neutral/disagree or low/moderate/high, according to the item. The mean “Palliative Scores” for the two scenarios were compared according to participants’ characteristics (medical vs. surgical staff, physicians vs. nurses, and senior vs. junior staff members) using *t*-test, and differences in median values were tested using the Wilcoxon non-parametric test. For responses to the PC perception items scored in three categories, one-way analysis of variance (ANOVA), and the nonparametric Kruskal–Wallis test were applied for the comparison of means and median “Palliative Scores” values of the two scenarios. In situations where the ANOVA or the Kruskal–Wallis tests came out statistically significant, multiple comparisons procedures using a post-hoc test were used to determine where the differences between the categories occur, and Bonferroni correction was applied. A value of *p* < 0.05 was considered statistically significant.

## 3. Results

### 3.1. Participants

As shown in Table 1, the sample included 315 health personnel: 159 physicians and 156 nurses. Median age was 33 years (mean 35.5 years, SD 9.4, range 24–71); 167 of them (53%) were females, half had four years or less of professional experience (mean 7.7, SD 8.6, median 4, range 1–50 years); less than a third (*n* = 97, 30.8%) were senior staff (defined as medical staff who completed all stages of their medical specialization or nurses in management positions). The majority were from medical wards (*n* = 190, 60.3%), and 39.7% (*n* = 125) were from surgical wards.

### 3.2. Preferred Medical Treatment as Shown via Two Case Scenarios

The mean Palliative Score of the preferred medical treatments was 9.92 (SD 4.29; range 1–17; median 10) for the first scenario (gastrointestinal bleeding) and 6.22 (SD 2.92; range 1–11; median 6) for the second scenario (pneumonia with respiratory failure). The percentage of respondents who chose the various medical treatments in the two case scenarios is presented in Table 2. The treatments most frequently chosen for scenario I and II, respectively, were laboratory tests (84.1%; 69.8%) and an intravenous fluid infusion (79.4%; 72.7%). Blood transfusion was chosen by 68.6% for scenario I, antimicrobial therapy by 72.4% for scenario II, and nasogastric tube insertion by 67.9% for scenario I. Analgesia was chosen by 50.5% and 62.2% for scenarios I and II, respectively, and sedation by 58.4% for scenario II (sedation was not a listed treatment option in scenario I). Subcutaneous infusion of fluids, urgent gastroscopy, intubation and mechanical ventilation, and central vein line insertion were less commonly selected. We found that 96% of the participants chose at least one alternative with the highest potential for causing suffering in scenario I, and 90% chose at least one alternative with either moderate or high potential for causing suffering in scenario II.

### 3.3. Preferred Medical Treatment as Shown via Two Case Scenarios 

Medical staff had a more PC approach than surgical staff in scenario I (gastrointestinal bleeding) (Palliative Score 9.1 ± 4.1 vs. 9.8 ± 4.4 for medical and surgical wards, respectively, *p* < 0.001) (Table 3). In scenario II (pneumonia with respiratory failure), there was no difference in the Palliative Score between medical and surgical staff. Physicians and nurses did not differ in their chosen Palliative Score in both scenarios. Senior staff had a more PC approach than junior staff in scenario II (pneumonia with respiratory failure) (Palliative Score 5.5 ± 3.1 vs. 6.6 ± 2.8 for senior staff and junior staff, respectively, *p* = 0.002), but not in scenario I (gastrointestinal bleeding).

### 3.4. Palliative Care Perceptions

Most participants (*n* = 284, 90.4%) agreed that PC is appropriate for patients with AD. On the other hand, their assessment regarding their colleagues’ acceptance of a PC approach was lower (73.6%) (Table 4). Approximately half of the participants reported feeling comfortable conducting end-of-life discussions (58.0%), having the ability to make care decisions (*n* = 178, 56.7%), and not being stressed or avoiding caring for a patient with AD (*n* = 165, 52.5%). About half (*n* = 146, 46.5%) disagreed that PC could expose them to a lawsuit. Most (*n* = 219, 69.7%) were concerned regarding the reaction of family members to the administration of PC.

### 3.5. Association between Palliative Care Perceptions and the Preferred Treatments 

A positive perception of PC as appropriate for patients with AD as perceived by colleagues was associated with fewer chosen aggressive medical treatment options. This association was statistically significant for scenario II (respiratory failure, *p* = 0.001 for mean and *p* = 0.002 for median), but not for scenario I (gastrointestinal bleeding, *p* = 0.1 and *p* = 0.7 for mean and median, respectively) (Table 4). 

Participants who agreed with feeling comfortable having end-of-life discussions in scenario II report less aggressive care (agree vs. neutral *p* < 0.001; agree vs. disagree *p* = 0.01 for comparison of means and agree vs. neutral for median, *p* = 0.001). Feeling guilty about the death of a patient with AD was associated with choice of more aggressive care treatments in scenario II; participants who disagreed had significantly lower mean scores than those who were neutral (*p* = 0.002) or agreed with this statement (*p* = 0.046). This pattern has also been seen when median scores were compared (6 and 7 for disagree and neutral, respectively, *p* = 0.009). The ability to make treatment decisions for patients with AD was related to the choice of less aggressive care treatment options in scenario I (agree vs. disagree, *p* = 0.03 for comparison of mean and median values). 

Participants who reported having lower feelings of stress and avoidance while caring for patients with AD chose less aggressive care treatments than those who were neutral or agreed that care of patients with AD causes stress and avoidance (*p* = 0.006 for overall comparison of mean values in scenario II). Agreement with the statement that PC exposes the healthcare provider to a lawsuit was associated with more aggressive treatment choices in scenario II (neutral vs. disagree, *p* = 0.006 for median). Higher levels of concern for PC lawsuits were associated with choice of more aggressive treatments in scenario II (high vs. low, *p* = 0.001 for comparison of mean and median).

We also observed an association between the concern regarding criticism by senior staff members and more aggressive treatment choices in both scenarios (Palliative Score 9.1 ± 4.6, 9.8 ± 4.0, and 10.8 ± 4.1 for low, neutral, and high levels of concern, *p* = 0.02 for comparison of means in scenario I and 5.0 ± 2.8, 6.5 ± 2.9, and 7.3 ± 2.7 for low, neutral, and high levels of concern, *p* < 0.001 for comparison of mean scores in scenario II). Statistically significant differences were also observed for low vs. high and low vs. neutral; (*p* < 0.001 and *p* = 0.009, respectively) for comparison of median scores in scenario II. Lack of organizational support was associated with choice of more aggressive care in scenario II (disagree vs. neutral, *p* = 0.009 for comparison of means, disagree vs. agree, *p* = 0.043, neutral vs. agree, *p* = 0.009 for comparison of median). Apprehension of the reaction of patients’ family members to PC was associated with choice of more aggressive care treatments in scenario II; high vs. low, *p* = 0.035 for comparison of mean.

## 4. Discussion

Our study was aimed at the assessment of the perception and implementation of concepts of PC and end-of-life decisions addressing patients with AD with life-threatening conditions. Our main findings were that while 90% of the study cohort of physicians and nurses accepted PC as appropriate for patients with AD, only half of them chose a PC approach (analgesia and sedation) as the treatment option in acute life-threatening conditions, while slightly more than a third considered invasive interventions, such as endoscopy in the case of gastrointestinal bleeding and intubation and mechanical ventilation in the case of respiratory failure. Nearly all participants chose at least one intervention with the highest potential for causing suffering in scenario I and at least one intervention with moderate or high potential for causing suffering in scenario II.

The most frequently chosen medical interventions were laboratory tests, intravenous fluids, blood transfusions, and antimicrobial therapy. These treatments may fall within a wide range of goals of care, including “basic care” provided to acute care patients as well as PC. Performing laboratory tests when there is a clinical necessity is consistent with both PC and curative care, since the test results might have implications for care and constitute minimal accompanying risk and discomfort [21]. While intravenous fluids may be a reasonable treatment for all care approaches (excluding the near-death period), there is no consensus on whether blood/blood products or antimicrobial therapy should be considered as a part of PC. A review of the literature revealed that four out of seven studies reported longer survival among patients in need of PC receiving antibacterial therapy, while the remaining studies found no difference in survival [22]. Although antimicrobial therapy has been used in PC to achieve symptom relief [23], some claim that when symptoms are absent, and the patient cannot swallow, antimicrobial therapy should not be considered palliative [24,25]. Nevertheless, studies have shown that most patients, their family members, and healthcare workers prefer antibiotics, even when the patient is terminally ill or suffering from AD [22,26,27]. Antimicrobial therapy is viewed as a low-burden intervention with the potential to treat reversible conditions that may be associated with suffering [22]. Similar perceptions have been found with respect to blood/blood products in PC. A systematic review found some short-term benefit with respect to symptom alleviation, and one study found prolonged survival with the supplementation of blood [28].

Only about half of the respondents selected interventions aimed at ameliorating suffering such as sedation and analgesia. This finding is surprising and likely reflects a lack of awareness of suffering in uncomplaining patients with AD. Some participants favored invasive interventions known to cause patients discomfort or suffering, such as insertion of a nasogastric tube, urgent gastroscopy, or placement of a central venous line. About one third selected intubation and mechanical ventilation, despite the fact that the Law of the Dying Patient in Israel permits withholding, but not withdrawing, continuous interventions such as mechanical ventilation, in patients with an estimated life expectancy of less than six months [29,30]. Studies have reported an increase in the use of mechanical ventilation among patients with dementia [31,32]. The usual life-saving orientation in acute care settings and the lack of palliative/geriatric medical training may explain increasing utilization of aggressive care and the relatively low rate of adopting a PC approach. This explanation is supported by a study among nurses and nurse-assistants indicating that knowledge and training in palliative and dementia care were associated with higher levels of positive attitudes toward PC in patients with dementia [33].

One possible explanation for the preference of aggressive, life-prolonging medical treatment in acute care settings among many of our study participants is the socialization process that shapes the perceptions of health personnel [34,35]. However, studies show that preference for more aggressive medical treatment does not lead to an improved prognosis or quality of life in patients with AD [36,37,38,39,40]. Providers appear to be skeptical about PC for patients with chronic diseases other than cancer [41], and many lack the self-confidence to provide such care [42]. A Finnish study concluded that the low rate of implementation of PC could be associated with unrecognized palliative needs of patients with dementia [43].

Medical staff chose less aggressive care than surgical staff in the case of gastrointestinal bleeding (scenario I). The surgical staff utilized more invasive interventions in this case (such as gastroscopy), likely reflecting their automatic response to gastrointestinal bleeding as a part of their professional training, regardless of the underlying state of AD, while the medical personnel preferred more conservative management. 

We interviewed a subset of the current sample (15 physicians and 11 nurses) in a qualitative study to investigate the healthcare personnel’s thinking processes associated with a case scenario of bowel obstruction related to a space-occupying lesion in a patient with AD. We found that surgical health personnel tended to focus on the immediate interventional response, while medical staff focused mainly on palliative measures [18]. Interestingly, there was no difference in the treatment preferences between medical and surgical disciplines in the case of pneumonia with respiratory failure, indicating that professional orientation was not a major determinant in this condition. In addition, both in the current study and in the qualitative study described here [18], there was no difference in the level of aggressive care between physicians and nurses. By contrast, other studies have reported a preference of nurses for PC, while physicians favored an aggressive approach, likely reflecting their own perception and training as life-savers [44,45,46,47]. We propose that our contradicting finding may reflect different perspectives in Israel regarding the role of nurses vs. physicians in critical medical decisions, especially under acute care settings.

The senior level staff had a more palliative approach than the junior staff in the case of pneumonia with respiratory failure. This is likely due to their experience regarding the long-term course, prognosis, and suffering of those patients, leading to their reluctance to initiate mechanical ventilation under these settings. Furthermore, senior staff are more experienced in end-of-life decisions and assumed to be less apprehensive in taking responsibility. On the other hand, there was no difference between the senior and junior staff in the approach to the case of acute gastrointestinal bleeding. This is most likely based on their assumption that this is a potentially reversible critical condition, unrelated to the end-stage cognitive impairment.

In our study, most of the respondents reported that PC was appropriate for AD patients, with a high percentage perceiving agreement among their colleagues. This finding suggests improved PC perceptions, corroborating findings of recent studies [48]. Noteworthy, however, a Finnish study, assessing the tendency of physicians to choose PC for patients with AD, using a hypothetical case scenario, reported that PC was chosen less frequently in 2015 than in 1999. The authors suggest that increased reports of legal concerns among physicians in 2015 may partially explain this shift of preferences [43].

We found an association between treatment choices and perceived level of apprehension regarding criticism by senior personnel in both hypothetical scenarios. An ability to make care decisions for patients with AD was associated with medical treatment choices in scenario I (gastrointestinal bleeding) but not in scenario II. By contrast, provider perceptions related to the other aspects of PC in patients with AD were found to be associated with medical treatment choices in scenario II (pneumonia with respiratory failure), but not in scenario I (gastrointestinal bleeding). This striking difference between the two scenarios may be due to the inherent nature of these medical conditions. Acute gastrointestinal bleeding is a potentially reversible problem not directly related to AD that can be successfully managed irrespective of the general PC approach. In contrast, pneumonia with respiratory failure is likely directly related to the underlying severe cognitive impairment and the associated neurologic functional deficits, with a poor anticipated immediate and long-term outcome [1]. Indeed, aspiration pneumonia is a leading terminal event in most AD patients. Furthermore, mechanical ventilation in these settings may turn out to be permanent in the case of weaning failure. This is a major ethical issue in Israel and in other societies where disconnecting a patient from mechanical ventilation is illegal or unacceptable [49]. 

Only about half of the respondents reported positive feelings about caring for patients with AD. Feeling comfortable with end-of-life discussions, lack of negative feelings, such as guilt, stress, and lack of desire to avoid being involved in the care of patients with AD, were all associated with favoring PC treatment choices in the case of pneumonia with respiratory failure. Negative self-perceptions about PC may be related to insufficient knowledge and experience in providing PC [50,51]. Indeed, the lack of acquaintance regarding the options for end-of-life care was found to be a barrier to effective discussions about end-of-life among internal medicine residents [52]. We have recently reported that health personnel in general hospitals report low rates of end of life discussions with family members of hospitalized patients with dementia [53]. Some investigators found that knowledge deficits were negatively correlated with perceived self-efficacy [54,55], and positively correlated with lack of confidence in making care decisions for patients with AD and concerns about their ability to provide quality end-of-life care [12,56]. Other researchers argue that negative emotions and stress arising from caring for the seriously ill patient might negatively impact quality of care and even lead to poor judgment and performance, and to incoherent care goals [13]. Stress is another factor that decision makers must face in most life-or-death situations [5]. 

Our findings demonstrate statistically significant associations between perceptions related to legal concerns and preferred medical treatments in scenario II (pneumonia with respiratory failure). Perceptions of risk and ability to cope may play a role in the care of patients with AD [16]. Substantial legal concerns were found to be associated with more aggressive treatment choices [43]. Others found critical gaps and a lack of knowledge of relevant legal status among health personnel [57], promoting dis-concordant perceptions and performance. True enough, Jox et al. [58] found that health care providers supporting PC acknowledged that choosing this approach may result in legal actions and disciplinary sanctions, and, therefore, imposing the provision of care they considered futile [58,59].

We found that about a third of our respondents were very concerned about senior-level criticism regarding PC decisions, and about a third agreed that there was a lack of organizational support for PC. Both items were associated with choosing more aggressive medical treatment. Evidence from previous studies supports these findings. Providers who consider care options that do not follow the perceived traditional curative care approach may be worried about a negative response from other staff members, leading to the suppression of the pursuit of initial treatment intentions [16,60,61]. This could be due to the organizational culture of acute care hospitals that encourages a curative treatment approach and uses mortality rates as an indicator of quality of care. This might be especially relevant to surgical disciplines. Therefore, when a junior health care worker considers PC, it might be perceived as different from the accepted care plan. In such cases, the junior practitioner will tend to consult with peers and senior staff. Ranse et al. [62] reported that informal collegial support may assist with the management of end-of-life care, whereas a lack of perceived organizational support in the context of care decisions for patients with life-limiting illness may be a barrier to PC [20]. 

More than two thirds of the participants in our study reported having a high level of perceived concern about the reaction of family members to PC. The higher the level of concern, the more aggressive the chosen medical treatments in scenario II (pneumonia with respiratory failure). Our findings are consistent with previous reports, showing that physicians are reluctant to offer PC when they anticipate that their recommendation would be misunderstood by the patient’s representative as giving up on patient’s care [41]. In a systematic review of barriers to prescribing PC in cases of life-limiting disease, an important factor was the perceptions of the patient’s family [63]. Often, family members coax health personnel to provide futile care to patients with AD. Medical staff may be intimidated and avoid PC to prevent conflict, even though it contradicts their own best judgment [58,63], leading to less appropriate care for the patient [64].

Although benefits and acceptance of PC have evolved over the last decades, in certain societies and under specific medical settings, such as in departments of emergency medicine, its implementation is limited [65]. Varied tools that screen for the presence of PC needs were employed and adjusted for emergency care [66]. However, perceptions of PC appropriate to patients with chronic medical conditions are seldom adopted in the presence of acute life-threatening illness in the emergency room [67].

This study has a few limitations. Possibly, the treatment options chosen by the participants in the described scenarios do not accurately reflect real-life responses. In addition, the study was carried out in two acute care tertiary hospitals in Israel, using local experts for questionnaire validation, and therefore, the generalization of the findings to primary care hospitals and to other countries and societies may not be applicable. Indeed, we did not investigate the possible impact of culture and religion on our findings. Staff perceptions about PC were focused on immediate decision-making encountered in acute care settings. Therefore, many aspects of PC such as a multidisciplinary approach, or response to emotional, spiritual, or social needs have not been addressed. The list of medical treatment options provided for both scenarios relates to the management of acute conditions and not to end-of-life care or to other interventions such as palliative sedation. In our study, we excluded physicians and nurses with former postgraduate training in geriatric and/or PC. Therefore, the impact of such training on perceptions about PC and treatment decisions in emergency situations for patients with AD was not investigated. Our questionnaire was content validated by Israeli physicians, not involved in this study, who were all experts in geriatric medicine and internal medicine. Further studies are necessary to validate our questionnaire in other countries and societies, addressing the cultural and legal differences regarding the attitude towards dementia and end-of-life care and management of patients with AD.

## 5. Conclusions

This study presents the perceptions of health personnel in the context of treating patients with AD under acute life-threatening medical situations. Although most participants expressed favored PC for this population, it is evident that when considering actual clinical decisions in these emergency situations, many barriers remain that impede the implementation of PC. This was demonstrated by a large percentage of respondents who did not choose PC treatments, as well as discomfort and concern expressed regarding related aspects that may affect them personally following the implementation of PC. Our findings could form a basis for the development of effective PC training programs for acute care personnel and for the revision of organizational norms of care for patients with AD. Acquainting emergency care practitioners with the benefits of PC for patients with AD may potentially impact their decisions when treating this population. Most impotent, adoption of advance directives, and discussions in advance with family members of patients with AD may prevent futile or unnecessary referrals to hospitals and may facilitate implementation of PC in this population during acute life-threatening illnesses.

## Figures and Tables

**Table 1 ijerph-19-10236-t001:** Demographic characteristics of the participants (*n* = 315).

**Age (years)**	Mean ± SD	35.5 ± 9.4
	Median	33.0
	Range	24–71
**Work experience (years)**	Mean ± SD	7.7 ± 8.6
	Median	4.0
	Range	1–50
**Gender**	Male *n* (%)	148 (47.0)
	Female *n* (%)	167 (53.0)
**Profession**	Physician *n* (%)	159 (50.4)
	Nurse *n* (%)	156 (49.6)
**Position**	Senior staff *n* (%)	97 (30.8)
	Junior staff *n* (%)	218 (69.2)
**Ward**	Medical ward *n* (%)	190 (60.3)
	Surgical ward *n* (%)	125 (39.7)

**Table 2 ijerph-19-10236-t002:** Selected treatment options for case scenarios I and II (*n* = 315).

	Scenario I Gastrointestinal Bleeding			Scenario II Pneumonia with Respiratory Failure		
	Item Palliative Score *	*n*	% Selected	Item Palliative Score *	*n*	% Selected
Intubation and mechanical ventilation	N.R.	N.R.	N.R.	3	117	37.1
Urgent gastroscopy	3	131	41.6	N.R.	N.R.	N.R.
Nasogastric tube insertion	3	214	67.9	N.R.	N.R.	N.R.
Central venous line insertion	3	132	41.9	N.R.	N.R.	N.R.
Blood transfusion	2	216	68.6	N.R.	N.R.	N.R.
Intravenous fluid infusion	2	250	79.4	2	229	72.7
Antimicrobial therapy	N.R.	N.R.	N.R.	2	228	72.4
Laboratory tests	2	265	84.1	2	220	69.8
Subcutaneous fluid infusion	1	67	21.3	N.R.	N.R.	N.R.
Analgesia	−1 **	159	50.5	−1 **	196	62.2
Sedation	N.R.	N.R.	N.R.	−1 **	184	58.4

N.R. not relevant; * Item score for each chosen intervention. The palliative score equals the sum of all of the items; ** Palliative measures chosen were given a (−1) score and +1 if not chosen.

**Table 3 ijerph-19-10236-t003:** Palliative Scores in the two hypothetical scenarios according to participants’ characteristics (*n* = 314).

			Palliative Score *			
			Scenario I Gastrointestinal Bleeding		Scenario II Pneumonia with Respiratory Failure	
		*n*	Mean ± SD	Median	Mean ± SD	Median
Ward	Medical wards	189	9.1 ± 4.1	9.0	6.2 ± 2.8	6.0
	Surgical wards	125	9.8 ± 4.4	12.0	6.3 ± 3.1	6.0
	*p*-value		<0.001	<0.001	0.8	0.6
Profession	Physician	158	9.9 ± 4.3	10.0	6.1 ± 2.9	6.0
	Nurse	156	10.0 ± 4.3	10.0	6.4 ± 3.0	6.0
	*p*-value		0.4	0.7	0.15	0.18
Position	Senior	96	9.5 ± 4.9	9.5	5.5 ± 3.1	6.0
	Junior	218	10.1 ± 4.0	10.0	6.6 ± 2.8	6.0
	*p*-value		0.2	0.9	0.002	0.07

* palliative score is the sum of the scores of all items (treatment alternatives) chosen for each scenario.

**Table 4 ijerph-19-10236-t004:** Association between palliative care perceptions and treatment approach in the two hypothetical scenarios (*n* = 314).

				Palliative Score *			
		All Participants		Scenario 1 Gastrointestinal Bleeding		Scenario II Pneumonia with Respiratory Failure	
		*n*	%	Mean ± SD	Median	Mean ± SD	Median
**PC appropriate for patients with AD**	**Positive:** Always/very often/often	284	90.4	9.8 ± 4.3	10.0	6.2 ± 2.9	6.0
	**Negative:** No/seldom	30	9.6	11.2 ± 3.9	11.5	6.7 ± 2.7	6.5
	***p*-value**			0.07	0.4	0.3	0.5
**Perceived PC appropriate for patients with AD by Colleagues**	**Positive:** Agree	231	73.6	9.7 ± 4.2	10.0	5.9 ± 2.9	6.0
	**Negative:** Neutral/disagree	83	26.4	10.5 ± 4.4	10.0	7.2 ± 2.8	7.0
	***p* Value**			0.1	0.7	0.001	0.002
**Comfortable with conducting end-of-life conversation with family/patient**	**Agree**	182	58.0	9.7 ± 4.5	10.0	5.6 ± 3.0	6.0
	**Neutral**	79	25.2	10.1 ± 3.8	10.0	7.1 ± 2.9	7.0
	**Disagree**	53	16.8	10.4 ± 4.1	12.0	6.9 ± 2.4	6.0
	***p* Value**			0.5	0.6	0.001	0.002
**I perceive a death of a patient with dementia as a therapeutic failure accompanied by a sense of guilt and stress**	**Disagree**	188	59.9	9.7 ± 4.6	9.0	5.7 ± 3.1	6.0
	**Neutral**	83	26.4	10.2 ± 3.8	10.0	7.1 ± 2.5	7.0
	**Agree**	43	13.7	10.5 ± 4.0	11.0	7.0 ± 2.4	7.0
	***p* Value**			0.4	0.07	<0.001	0.005
**Ability to make care decisions for patients with AD**	**Agree**	178	56.7	9.4 ± 4.2	9.0	6.0 ± 2.8	6.0
	**Neutral**	77	24.5	10.4 ± 4.2	10.0	6.7 ± 3.3	7.0
	**Disagree**	59	18.8	11.0 ± 4.3	12.0	6.2 ± 2.8	6.0
	***p* Value**			0.03	0.03	0.3	0.2
**Care of patients with AD makes me feel stressed and want to avoid making care decisions**	**Disagree**	165	52.5	9.7 ± 4.5	10.0	5.8 ± 3.0	6.0
	**Neutral**	72	22.9	10.8 ± 3.9	11.0	7.2 ± 2.2	7.0
	**Agree**	77	24.5	9.6 ± 4.1	10.0	6.2 ± 3.2	6.0
	***p* value**			0.2	0.04	0.006	0.06
**PC exposes the healthcare provider to a lawsuit**	**Disagree**	146	46.5	9.9 ± 4.3	10.0	5.9 ± 2.8	6.0
	**Neutral**	100	31.8	9.9 ± 4.4	10.0	6.5 ± 2.9	7.0
	**Agree**	68	21.7	9.9 ± 4.1	10.0	6.6 ± 3.0	6.0
	***p* value**			0.9	0.9	0.12	0.006
**Apprehension of a lawsuit following PC decisions**	**Low**	104	33.1.	9.3 ± 4.5	9.0	5.4 ± 2.8	6.0
	**Neutral**	81	25.8	9.9 ± 4.4	10.0	6.2 ± 2.8	6.0
	**High**	129	33.1	10.5 ± 4.0	10.0	6.9 ± 2.9	7.0
	***p* value**			0.09	0.3	0.001	0.001
**Apprehension of senior-level response to PC decisions**	**Low**	109	34.7	9.1 ± 4.6	9.0	5.0 ± 2.8	6.0
	**Neutral**	99	31.7	9.8 ± 4.0	10.0	6.5 ± 2.9	6.0
	**High**	106	33.8.	10.8 ± 4.1	11.0	7.3 ± 2.7	7.5
	***p* value**			0.02	0.2	<0.001	0.001
**Lack of organizational support for PC decisions for patients with AD**	**Disagree**	123	39.2	9.7 ± 4.4	10.0	5.7 ± 2.7	6.0
	**Neutral**	106	33.8	10.0 ± 4.2	10.0	6.7 ± 2.9	7.0
	**Agree**	85	27.1	10.2 ± 4.2	10.0	6.5 ± 3.1	6.0
	***p* value**			0.7	0.6	0.03	0.006
**To what extent does apprehension of family response influence care decisions?**	**Low**	41	13.1	8.6 ± 4.7	9.0	5.2 ± 3.1	6.0
	**Neutral**	54	17.2	9.9 ± 4.9	10.0	5.9 ± 2.7	6.0
	**High**	219	69.7	10.2 ± 4.0	10.0	6.5 ± 2.9	6.0
	***p* value**			0.08	0.3	0.03	0.06

AD, advanced dementia; PC, palliative care; * palliative score is the sum of the scores of all items (treatment alternatives) chosen for each scenario.

## Data Availability

The data presented in this study are available on request from the corresponding author. The data are not publicly available due to ethical reasons.

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
