# Peer review of "Do Perceptions about Palliative Care Affect Emergency Decisions of Health Personnel for Patients with Advanced Dementia?"

_ijerph, 2022, doi:10.3390/ijerph191610236_

Round 1

Reviewer 1 Report

This study aimed to examine the preferred medical treatment in emergency situations for patients with advanced dementia and its association with perceptions of palliative care. The research is relevant, original and offers a contribution to the knowledge of the area. In that sense, I congratulate the authors. The manuscript, however, needs to be revised and some suggestions and questions are presented here for good application of findings.

 Title:

1. The title is fine, but I am not sure if it is attractive to catch readers’ attention. Please try to find another title.

 Abstract:

2. The abstract is well done. However, it seems to me that all information and results are presented in the abstract section, reducing the focus on the main finding. I suggest rewriting part of the abstract. Furthermore, abbreviations should be avoid in this section.

 Keywords:

3. Use keywords according to MESH (Medical Subject Headings) terms.

 Introduction:

4. Avoid using too many abbreviations. In the first paragraph, for example, there are 3 different abbreviations.

5. Write smaller paragraphs. The introduction content is correct, but smaller paragraphs are preferred in scientific English.

6. Avoid using too many references in small sentences. For example, rows 70-74 you cited 9 references.

 Methods:

7. Detail more inclusion and exclusion criteria.

8. Explain why participants should not have “training in geriatric medicine and/or Palliative Care”.

9. I am not convinced that “Six board-certified geriatric physicians working in acute care hospitals validated the content of the questionnaire” is enough to guarantee the quality of the questionnaire. Wasn’t easier to use a validated instrument?  This is to me the key point of the study that can invalidate the findings.

 Results:

10. Clarify to readers “Score if selected” (table 2). What does this score means? What does a negative value mean?

11. I’m not sure what does the scores in table 4 means, in a created instrument not previously validated.

12. Paragraph after item 3.4. is too long and it should be split into different paragraphs.

 Discussions:

13. Several questions that I have highlighted previously should be included in the discussion section.

14. In the methods section you said that this is a mixed-methods (qualitative/quantitative) research. What was the qualitative aspect of the research?

 References:

15. Please avoid self-citation. It is not wrong to cite the studies developed by the research group. However, this practice should be restricted to few articles.

Reviewer 2 Report

Reviewer feedback: Recommended minor revision

Title: Emergency Decisions of Health Care Workers Regarding Patients with Advanced Dementia: Association with Perceptions About Palliative Care 

Dear Authors. Thanks for your study. Please, read the review feedback here:

This is a relevant and current topic that will be of interest to the readers of this journal. The research is original and important in the field of health care for persons with dementia.

The aim of the study was to examine the preferred medical treatment in emergency situations for patients with advanced dementia and its association with perceptions of palliative care.

The study followed the guidelines for research ethics and was approved by the ethics committee of each hospital. Participants received oral and written information about the study and provided their written consent. Could you please provide the ethical information (including the Ethics Committee and the approval number)?

Materials and Methods are well-structured, but I would appreciate information on the data collection period. How can the authors prove that the participants in this study voluntarily completed the questionnaire during regular nursing or medical staff meetings? Please, add more information. 

The discussion reflected on findings of the assessment of the perception and implementation of concepts of palliative care and end-of-life decisions addressing patients with AD with life-threatening conditions. The main findings were that while 90% of the study cohort of physicians and nurses accepted palliative care as appropriate for patients with advanced dementia, only half of them chose the palliative care approach.

I appreciate that the authors in the conclusion suggested concrete measures for educators such as palliative care training programs for acute care personnel and for leadership and professionals such as the revision of organizational norms of care for patients with advanced dementia.

The article is clear and well-structured. Its individual parts are interlinked to form a logical whole - introduction, methods, results, discussion, and conclusion. The length of the abstract is appreciated and contains all necessary findings. The Introduction is clear and well organised. I appreciate that the authors explained the main terms and situations for patients with advanced dementia and palliative care.

The manuscript is written in English that is easy to understand. Once corrections Materials and Methods, the paper would be most suitable for publication.

Dear authors, congratulations on your research - good luck!

Oslo, Norway, 29th of July 2022

Round 2

Reviewer 1 Report

Dear authors. Thank you for addressing all my comments. I saw an inprovement in this new version of the manuscript.